# The Effect of Illumination Patterns during Mung Bean Seed Germination on the Metabolite Composition of the Sprouts

**DOI:** 10.3390/plants12213772

**Published:** 2023-11-04

**Authors:** Irina N. Perchuk, Tatyana. V. Shelenga, Marina. O. Burlyaeva

**Affiliations:** N.I. Vavilov All-Russian Institute of Plant Genetic Resources, 42,44, B. Morskaya Street, 190000 Saint-Petersburg, Russia; t.shelenga@vir.nw.ru

**Keywords:** mung bean, *Vigna radiata* (L.) Wilczek, sprouts, leaves, metabolomic profile, illumination patterns during germination

## Abstract

Mung bean (*Vigna radiata* (L.) Wilczek) sprouts are popular over the world because of their taste, nutritional value, well-balanced biochemical composition, and other properties beneficial for human health. Germination conditions affect the composition of metabolites in mung bean sprouts, so a detailed study into its variability is required. This article presents the results of a comparison of the metabolite composition in the leaves of mung bean sprouts germinated first in the dark (DS) and then in the light (LS). Gas chromatography with mass spectrometry (GC–MS) made it possible to identify more than 100 compounds representing various groups of phytochemicals. Alcohols, amino acids, and saccharides predominated in the total amount of compounds. The analysis of metabolomic profiles exposed a fairly high intra- and intervarietal variability in the metabolite content. DS and LS differed in the qualitative and quantitative content of the identified compounds. The intravarietal variability was more pronounced in DS than in LS. DS demonstrated higher levels of saccharides, fatty acids, acylglycerols, and phenolic compounds, while amino acids were higher in LS. Changes were recorded in the quantitative content of metabolites participating in the response of plants to stressors—ornithine, proline, GABA, inositol derivatives, etc. The changes were probably induced by the stress experienced by the sprouts when they were transferred from shade to light. The analysis of variance and principal factor analysis showed the statistically significant effect of germination conditions on the content of individual compounds in leaves. The identified features of metabolite variability in mung bean genotypes grown under different conditions will contribute to more accurate selection of an illumination pattern to obtain sprouts with desirable biochemical compositions for use in various diets and products with high nutritional value.

## 1. Introduction

The issue of health-friendly nutrition will always remain pertinent to humankind. Plants are one of the sources of compounds necessary for a normal human life. In addition, plants have been used as medications and means of preventing various diseases since ancient times. In the most recent decades, the problem of optimal plant genetic resources utilization has been under special scrutiny, and research efforts have abruptly intensified in this sphere. At the same time, with the development and promotion of the metabolomic approach and related technologies, the study of plants has advanced to a new level [1]. This is especially important for food such as plant sprouts, since their germination is accompanied by the activation of all metabolic processes. Along with the hydrolysis of high-molecular compounds reserved in seeds, there is the synthesis of new bioactive compounds, ensuring normal plant growth and development and the protection of sprouts against various stressors.

Modern methods of analysis, such as gas chromatography (GC) and high-performance liquid chromatography (HPLC) combined with mass spectrometry (MS), are widely applied to analyze sprouts of various crops, including legumes, as they provide rapid access to substantial information about their metabolite composition and its changes during germination [1,2,3,4,5]. Studying the effect of sprouting on the nutritional value of various legume (common bean, soybean, lentil, pea, cowpeas, and mung bean) seeds shows that sprouts contain more free amino acids, minerals, dietary fibers, and phenolic compounds than seeds. In addition, sprouts have lower total carbohydrate and fat content levels and fewer antinutrients. Sprouting affects various types of biological activity in seeds [6,7,8,9,10,11,12,13,14,15,16,17]. Thus, sprouting is believed to improve nutritional qualities and the potential health benefits of seeds.

The data accumulated recently testify that the quality of legume sprouts—the composition of their metabolites—largely depends on the plant species and genotype and the “age” of the sprouts. It may be explained by the fact that the seed metabolism is so dynamic during germination that the changes in the content of different metabolite groups are observed immediately in the first hours of germination and continue throughout the entire period of sprout development [2,3,6,11,12,18,19,20,21]. Germination conditions also have a significant impact on the biochemical composition in sprouts. The use of elicitors (ethylene; glutamic, ascorbic, and folic acids; lactoferrin; fish protein hydrolysate; oregano extract, etc.) helps to increase the antioxidant and antimicrobial activity of legume sprouts [22,23,24,25]. The protein content in sprouts can be affected by the temperature [26]. Illumination patterns during germination play an important role for certain groups of metabolites. Sprouts of the same plant accession grown in the dark or under the light differ in their biochemical composition. A number of studies demonstrated the influence of the light spectrum parameters (wavelength), various illumination patterns, and their combinations (for example, LED and UV) on the synthesis and accumulation of total phenolic compounds (PCs) in legume sprouts or individual phenolic acids, flavonoids, etc. Light also affects the accumulation of bioactive compounds such as amino acids, vitamin C, vitamin E, carotenoids, and photosynthetic pigments [8,17,27,28,29,30,31,32,33].

Mung bean (*Vigna radiata* (L.) Wilczek) is a crop commonly grown in countries with tropical and subtropical climates. The seeds of green pods are widely used in Asian, European, and American cuisines. Whole mung bean sprouts or their individual parts, in particular, their leaves, are eaten fresh. A significant number of compounds with various biological activities (antioxidant, antidiabetic, anti-inflammatory, antimicrobial, antiviral, anti-anemic, and antitumor) have been identified in mung bean seeds and sprouts. These compounds also exhibit antihyperlipidemic, antihypertensive, and antimutagenic properties [18,19,22,34,35,36,37,38,39]. Mung bean sprouts, due to their medicinal properties, easy digestibility, and nutritional value, are included in various diets, being used both for functional and disease-preventing nutrition. Previously, it was mentioned that the biochemical composition of seeds changes significantly during their germination. Proteolysis of mung bean seed proteins leads to the emergence of a large number of new peptides, with potential biological activity in some of them. For example, vicilin, a storage protein in mung bean that can provoke an allergic reaction in those who are susceptible, loses more than 70% of its immunoreactivity during seed sprouting. According to the authors, these results suggest that sprouting is a possible way to produce hypoallergenic food [4,40]. Compared to mature seeds, a multifold increase (two to nine times) is observed in the content of both the total PCs and individual PC groups, as well as a change in their qualitative composition. Depending on the germination schedule, the content of antioxidants such as vitamin C (3 to 24 times), vitamin E, and glutathione increases. Changes are recorded in the qualitative and quantitative composition of organic and amino acids, lipids, carbohydrates, etc. [2,3,11,19,20,41,42]. Sprouting significantly affects the content of the so-called antinutrient compounds—phytic acid, hemagglutinin, protease inhibitors, and tannins. For example, a reduction in the content of phytates can reach 15–76%. This improves the availability and digestibility of mung bean proteins and increases the content and bioavailability of Zn, Fe, and Ca, since phytic acid is able to bind these minerals and build up complexes with proteins. An increase in the content of other minerals (Na, K, P, Mg, and Mn) has also been registered. The process of mung bean seed germination is accompanied by a constant decrease in the content (leading to a complete absence) of the raffinose and stachyose oligosaccharides associated with flatulence [6,39,43,44,45,46,47].

The ever-growing scientific interest in mung bean sprouts is explained by the fact that they are one of the most widespread and popular legume food products among consumers. Hence, special attention is paid during their production to the development of optimal conditions for the accumulation of health-friendly metabolites, including the selection of the illumination pattern most favorable for the nutritional value of sprouts and their biochemical composition. Numerous studies on bioactive compounds in mung bean sprouts and the effect of germination conditions on their composition have been undertaken in recent years, but there is still very little data on the effect of light on the phytonutrient content in the sprout leaves. The plant exists as a sprout for a short period of time, and metabolic processes are very intense during this interval. Leaf metabolism is closely associated with photosynthesis, so it seems interesting how quickly the metabolite composition in sprout leaves can change when a sprout is removed from the conditions where no photosynthesis is induced (dark) and placed under the conditions where photosynthesis is active (light). The objective of our study was as follows: (1) to analyze the nutritional value (metabolite composition) of mung bean sprout leaves using the GC–MS technique; (2) to assess the effect of light on the variability of the metabolite composition in mung bean sprout leaves under changing illumination patterns, namely, when the sprouts are transferred from darkness to light; and (3) to quantify the statistical significance of the effect of accession (genotype) on the variability of the identified metabolites.

## 2. Results and Discussion

### 2.1. Metabolomic Profile Analysis

The metabolomic profiles of the leaves collected from the studied mung bean sprouts were observed to contain more than 100 compounds representing various plant metabolite classes (Appendix A). Their content in individual accessions varied significantly. Alcohols, saccharides, and amino acids dominated in the leaves of both groups of sprouts (light-exposed samples, or LS, and dark-exposed samples, or DS) (Table 1).

#### 2.1.1. Alcohols

Values of the total alcohol content in the studied accessions are presented in Table 1. A total of 15 alcohols and their derivatives were identified; almost all of them were sugar alcohols (Appendix A). On average, less than 1% of the total alcohol content was represented on aggregate in both DS and LS groups by ethanolamine, phytol, and *α*-tocopherol. Ethanolamine was identified in all sprouts. Phytol, whose residue is part of the chlorophylls, was only identified in LS, but its share in the total alcohol amount was extremely small: 0.1–0.6%. Trace amounts of *α*-tocopherol were only observed in three LS samples.

Among the identified sugar alcohols, threitol, erythritol, deoxyglucitol, sorbitol, and galactinols were only present in LS, with their share being less than 0.3%. In the DS and LS groups, as a whole, the total proportion of glycerol and glycerol-3-phosphate averaged 6.1 and 2.1%, respectively. However, it should be mentioned that in two sprouts of k-14407 from the DS group, this proportion reached 20% (in the others, it did not exceed 8%). The main part of the alcohols consisted of inositol and its derivatives—on average, 93% for DS and 95% for LS. Inositol naturally has nine isomers, some of which are found in plants. Identified in the leaves of mung bean sprouts were myo-, methyl-, and chiro-inositol, ononitol, and, in a single plant, allo-inositol. Among all isomers, only myo-inositol is synthesized de novo in plants and gives rise to its other derivatives. The activity of the key enzyme in the myo-inositol synthesis, myo-inositol-1L-phosphate synthase (EC 5.5.1.4), which catalyzes the conversion of D-glucose-6-phosphate to 1-L-myo-inositol-1-phosphate, was observed in the plastid, chloroplast, and cytosolic cell fractions of mung bean sprouts [48].

Myo-inositol plays an important role in plant life, participating in various processes of metabolism in plants. These include energy metabolism, embryogenesis, and synthesis of compounds in the phospholipids of chloroplast membranes (phosphoinositides) and cell walls (oligo- and polysaccharides such as raffinose and hemicellulose). Myo-inositol derivatives are important secondary messengers for various signaling in plants. In addition, myo-inositol is involved in plant growth stimulation by making up complexes with auxins, which explains its presence in the free form in sprouts. Phosphorylated derivatives of myo-inositol and phytic acid and its salts are a source of phosphorus in plants [49,50,51].

Average values of the total content of various inositol forms did not differ significantly in DS and LS (5924 ± 3039 and 6098 ± 2462 ppm, respectively); however, differences in their relative content were observed (Figure 1).

The proportion of methyl-inositol in DS leaves was higher than 50% of the total alcohol amount, but the share of myo-inositol was also quite high (36%). After the sprouts were transferred to the light, the proportion of methyl-inositol in leaves significantly increased (86%), while the percentage of myo-inositol accordingly decreased (2%). Such a significant increase in methyl-inositol content may be a response of plants to the stress caused by a change in the illumination conditions during their growth. It should be noted that a similar reaction was observed in the sprouts of all three accessions. According to published data, it is the methyl derivatives of myo-inositol that are involved in the response of plants to biotic and abiotic stressors [49].

#### 2.1.2. Saccharides

The identified saccharides were represented by mono- (pentoses and hexoses), di- (sucrose and rutinose), and trisaccharides (raffinose), and their derivatives, 16 compounds in total (Appendix A). For all three accessions as a whole and for each of them individually, the DS group exceeded the LS group in absolute saccharide content values (Table 1). The intravarietal variability of this character was more significant in the DS group than in LS. The DS group showed a higher proportion of disaccharides, represented mainly by sucrose. There were more monosaccharides in the LS group, with a dominance of mannose, galactose, and glucose with its derivatives (Figure 2a,b).

It may be associated with the fact that initially, while germinating in the dark, the sprouts received saccharides during the hydrolysis of seed carbohydrate reserves whose share can exceed 60%, with a significant part being starch [38,52]. The transfer to the light initiated an active process of photosynthesis in the leaves, the primary products of which were monosaccharides, so their share increased. A reduction in the total saccharide content during the transition to the light can be explained by the increased metabolism in sprouts associated with their ontogenesis, and the utilization of saccharides as a source of energy. The content of various saccharide groups in mung bean sprouts was variable depending on the conditions of seed germination: initially, the levels of individual saccharides in the germinating material abruptly increased, with a consequent decrease over time. Raffinose, stachyose, and verbascose were almost completely hydrolyzed already on the fifth day of germination [3,18,53]. In this study, the share of raffinose in sprouts averaged 0.2% (DS) and 0.6% (LS) (Appendix A).

#### 2.1.3. Free Amino Acids

About 30 free amino acids and their derivatives were identified in the leaves of DS and LS samples, including 8 essential and 9 non-proteinogenic ones (Appendix A). Each accession and all of them together in the LS group exceeded DS by 2–4.5 in the average total amino acid content (ppm) (Table 1). There were no significant differences in the qualitative composition for both groups of leaf samples. However, in relative units (shares of individual amino acids in their total amount), differences between the groups were registered. The LS group had a higher proportion of phenylalanine, asparagine, tyrosine, and ornithine + ornithine lactam (Figure 3). An increase in the content of phenylalanine in mung bean sprouts, as well as in lentil and pea sprouts, was observed by other researchers [8,42]. The proportion of asparagine, one of the transport forms of nitrogen in plants, was 29% in the LS group—the highest value among all amino acids (Appendix A). The DS group had more valine, aspartic acid, glutamic acid, leucine + isoleucine, proline, and oxoproline. These results are consistent with the data of other authors [17,21].

An increase in proline content during germination both in the dark and under the light was recorded earlier for other legumes [8]. Proline, along with ornithine, is involved in the plant’s response to environmental stressors. Through a series of enzymatic transformations, proline can be a source of hydroxyproline and glutamic acid in one synthetic pathway, and ornithine in another. At the same time, the reactions leading from proline to glutamic acid or ornithine are reversible [54,55,56]. In this study, the DS group showed a higher total proportion of proline, oxoproline, and glutamic acid (17.9%) than the LS group (9.3%). The LS group contained more ornithine and its derivative, ornithine lactam (18.6%), than DS (8.4%) (Appendix A). The amino acid citrulline was identified in the studied sprouts. Citrulline is not found in proteins; it is an intermediate product in the synthesis of such an important amino acid as arginine. The relative combined content of the citrulline and arginine averaged 2.2% in DS and 3.6% in LS. Quantitatively, however, their content was highly variable: 68 to 347 ppm in LS and 0 to 56 ppm in DS (Appendix A). It should be mentioned that citrulline and arginine are also involved in the response of plants to various stressors. Arginine is a possible precursor of the multifunctional nitric oxide signaling molecule NO and, together with ornithine, participates in the synthesis of polyamines. Citrulline is an effective hydroxyl radical scavenger [55,57,58,59]. It is interesting to note that at least some of the metabolic pathways of arginine, citrulline, ornithine, and glutamic acid are associated with chloroplasts (plastids)—cellular structures actively functioning in the light [57,59]. Various plant stresses are also reflected in the metabolism of *γ*-aminobutyric acid (GABA) [60]. The GABA content was higher in the LS group than in DS: 25–142 and 4–37 ppm, respectively. An increase in the GABA content during germination was observed in the mung bean and other legumes by a number of researchers [8,18]. In general, the LS group differed from the DS group in a higher total percentage of non-proteinogenic amino acids (hydroxy- and hydroxyproline, *β*-alanine, *β*-phenylalanine, *γ*-aminobutyric, pipecolic and hydroxypipecolic acids, ornithine, and ornithine lactam): 21% vs. 12%. Non-proteinogenic amino acids perform various functions in plants, including protection from environmental stressors [61].

#### 2.1.4. Organic Acids and Phosphoric Acid

In total, 33 organic acids and their derivatives were identified in the metabolomic profiles, and 24 of them were present in both groups of sprouts (Appendix A). If we regard each of the DS and LS groups as a whole, there are no significant differences between them either in the average total content of organic acids or the range of their variability. These characters for DS and LS were 1873 ± 1124 and 1754 ± 1268 ppm (Table 1), respectively. However, the tested accessions reacted differently to changing growing conditions. The transfer of k-14407 sprouts from shade to light led to a decrease in the total content of organic acids by more than 60%, but k-14408 and k-14416 showed a slight increase (Table 1). The acids involved in the basic energy metabolism processes or associated with them (oxalic, lactic, pyruvic, citric, succinic, fumaric, and malic) were represented better than others. The total proportions of the Krebs cycle acids amounted to 44% in DS and 76% in LS with reference to the total number of the identified acids. A significant share was represented by malic and citric acids. The proportion of malic acid averaged 30% in the DS group and 45% in LS. The content of citric acid was five times higher in LS (Figure 4a). A number of authors also reported significant malic acid levels in mung bean sprouts, noting that changes in the content of some organic acids (malic, citric, lactic, and fumaric) in the process of seed germination were not uniform [3,62].

In this study, the DS group differed from the LS group in a higher total proportion of aldonic acids (glyceric, threonic, ribonic, gluconic, etc.) and their derivatives—27% and 4%, respectively (Figure 4a). The content of phosphates in both groups of sprouts was comparable with the content of organic acids (Table 1).

#### 2.1.5. Lipids

Thirteen free fatty acids (FAs) and their derivatives, as well as monoacylglycerols (MAGs) of palmitic, stearic, and linoleic FAs, were identified in the leaves of mung bean sprouts (Appendix A). Among the saturated FAs, the main acids were palmitic and stearic, and among the unsaturated FAs, the main acids were oleic, linoleic, and linolenic. Major FAs among those identified in the studied plant material are involved in the synthesis of galactolipids, whose share reaches 80% of the total lipids in photosynthetic membranes of plant cells [63,64]. The dominance of the same acids in mung bean sprouts was also observed by other researchers [17,38,65]. The DS group as a whole and the DS of each accession individually were characterized by a higher amount of free FAs (Table 1). The ratio of saturated/unsaturated FAs was practically the same for DS and LS: 1.3 and 1.2, respectively. There were also no significant differences between the groups in the relative content of the major FAs (Figure 4b).

On the whole, according to publications, an increase in the content of free FAs was observed in mung bean seeds during their germination. However, this increase was uneven: the highest levels of free FAs were registered in the first 24 h of mung bean seed germination, but afterwards, a significant decrease was observed over several days, followed again by a slight increase [3,65]. E-S.A. Abdel-Rahman marked the dominance of unsaturated FAs in germinated seeds compared to dry ones [65]. P.V. Hung noticed that the exposure to light considerably increased the amount of unsaturated FAs and reduced the content of saturated FAs in the sprouts of mung bean compared to the seeds [17]. The opposite tendency was observed in mung bean sprouts germinated in the dark. It should be taken into account that the above-mentioned studies dealt with whole germinating seeds and not their separate parts.

An increase in the content of monoacylglycerols during seed germination was recorded in a number of works [65]. We identified monoacylglycerols (MAGs) of palmitic and stearic acids in all studied leaf samples in the amounts comparable to organic acids and free FAs. On the whole, the content of these compounds was higher in DS. However, sprouts of different mung bean accessions reacted differently to the changes in germination conditions. If the total MAG content in the LS of k-14408 and k-14416 decreased by 15–40% compared to their DS, the LS of k-14407 demonstrated a very abrupt (dramatic) decrease, down to trace amounts in some sprouts (Table 1). This phenomenon may be explained by the peculiarities of the metabolism in the said accession. We observed in this study a simultaneous decrease in the content of some FAs and monoglycerols in LS leaves. It is possible that when mung bean sprouts were exposed to light, the synthesis of new lipids for various cell biomembranes was activated in them, on the one hand, and the already existing membranes were rearranged (remodeled) in response to the stress experienced by plants, on the other hand. This is especially true for photosynthetic membranes, a significant share of which is concentrated in the leaves [64].

#### 2.1.6. Phytosterols

Five phytosterols (PSs) were identified in the leaves of DS and LS (Appendix A). The major ones were stigmasterol, *β*-sitosterol, and campesterol. The same phytosterols also predominated in the leaves of other legumes [66]. On the whole, the DS group contained slightly more PSs than the LS group. The transfer to a different illumination pattern had no effect on the total PS content in the sprouts of k-14408 and k-14416. However, the total PS content in the sprouts of k-14407 decreased by more than 60% (Table 1). In relative units with reference to the total PS amount in each accession and the entire set as a whole, *β*-sitosterol (57%) dominated in the DS group and stigmasterol (74%) in the LS group (Figure 5, Appendix A). Since *β*-sitosterol is the precursor in the synthesis of stigmasterol, such a change in the ratio of these major PSs could supposedly be a response of sprouts to the stress they experienced when the illumination conditions of germination changed.

When plants adapt to various stresses, an important role is assigned to the state of cell membranes. PSs, as structural components of cell membranes, stabilize them and participate in the regulation of their fluidity, permeability, and other processes of membrane metabolism. Free sterols are precursors of the bound (conjugated) sterols involved in the response of plants to biotic and abiotic stressors, and such bioactive compounds as steroidal saponins, steroidal glycoalkaloids, phytoecdysteroids, etc. [67,68]. PSs are also precursors of phytohormones engaged in plant development.

#### 2.1.7. Phenolic Compounds

An important class of secondary plant metabolites is formed by phenolic compounds (PCs) (Appendix A). They protect the plant from biotic and abiotic stressors and act as intermediates in biosynthesis processes and as structural elements in cell membranes, etc. Pyrogallol and phenolic acids (PAs) were identified in the studied accessions, including caffeic, 4-hydroxycinnamic, 2,3-dihydroxybenzoic, and benzoic acids. Pyrogallol in the amount of 1–9 ppm was found mainly in the LS group. Caffeic acid was present in all sprouts. Its content varied in DS and LS between 37–333 ppm and 11–95 ppm, respectively. Some authors marked this acid as dominating in mung bean sprouts [18,20,69]. In all DS and the LS of k-14408 and k-14416, the 4-hydroxycinnamic acid content varied between 13–148 ppm and 5–40 ppm, respectively. This metabolite was not identified in the LS of k-14407. The highest content of benzoic acid was observed in the LS sample of k-14408 (25–56 ppm). 2,3-dihydroxybenzoic acid was found only in three sprouts of k-14408 (Appendix A). Intra- and intervarietal variability of quantitative PC parameters were observed in the studied material. However, the same reaction to the change in illumination conditions was registered for all three accessions: a slight decrease in the PC content in mung bean sprouts when they were exposed to light (Table 1). The relative content of major phenolic acids in the sprout accessions is shown in Figure 6.

According to published data, various methods of analysis helped to identify a fairly large diversity of PAs in mung bean sprouts: caffeic, para-coumaric, ferulic, cinnamic, synapic, para-hydroxybenzoic, gallic, vanillic, syringic, chlorogenic, etc. PAs can be either free or bound, i.e., forming part of more complex PCs [18,19,21,38,69,70].

Mung bean sprouts also contain PC subclasses such as flavonoids [16,20,38,39]. The PC composition and content change in the process of germination, i.e., they depend on the “age” of a sprout [11,12,71]. It should be noted that most studies analyze the biochemical composition of the whole sprout and not its individual parts. A significant part of mung bean flavonoids (sometimes more than 90%) can be found in seed coats [39,72]. Extraction techniques and methods of analysis play an important role [35]. We used only an organic solvent, methanol, as an extractant, without further hydrolysis of the extract. Flavonoids in the free state were not identified in the leaf blades of mung bean sprouts.

### 2.2. Statistical Analysis

#### 2.2.1. ANOVA Results

##### The Effect of the Germination Conditions on the Metabolite Content Variability of Mung Bean Sprouts

A one-way analysis of variance (ANOVA) was carried out to verify the statistical significance of the associations between growth conditions of the tested sprouts and the variability in the content of biochemical compounds in them. The illumination pattern significantly influenced the variability in the content of lactic (the effect size percentage was 24.7%), pyruvic (16.6), oxalic (46.5), succinic (60.9), citric (37.5), tartaric (21.2), 3-hydroxypropionic (26.2), 4-hydroxycinnamic (25.4), glyceric (84.5), methylglyceric (14.7), ribonic (15.4), threonic (78.1), total lactones of threonic and erythronic (67.5), gluconic (47.1), and 6-phosphogluconic (43.1) acids; phosphate and methyl phosphate (14.9); methionine (41.2); threonine (56.8); phenylalanine (45.0); tryptophan (27.2); glycine (25.0); tyrosine (21.4); the sum of histidine and lysine (70.6); aspartic acid (16.1); asparagine (57.1); glutamic acid (58.3); glutamine (47.9); the sum of citrulline and arginine (56.4); *β*-alanine (22.4); proline (33.9); GABA (63.8); ornithine (50.7); *β*-phenyl-*α*-alanine (19.4); ornithine lactam (36.6); glycerol (37.5); the sum of triethol and erythritol (26.1); arabinitol (33.3); glycerol-3-phosphate (24.3); deoxyglucitol (76.2); sorbitol (35.5); methyl-inositol (19.4); ononitol (26.5); myo-inositol (71.9); galactinol (18.6); phytol (75.5); fructose (14.7); glucose (33.7); sucrose (48.1); the sum of mannose and galactose (31.0); the sum of rutinose and its derivatives (20.1); stigmasterol (20.2); *β*-sitosterol (66.4); C16:0 (28.2); C18:0 (36.4); C20:0 (24.6); C18:1 (17.6); C18:2 (43.1); OH28:0 (45.4); acylglycerols (16.4); pyrogallol (39.1); urea (43.6); and adenosine (21.6) (Appendix A). Growth conditions had the greatest effect on the variations in glyceric and threonic acids, deoxyglucitol, and myo-inositol. The effect size of their impact on the content of some metabolites involved in this or in the response of plants to stressors (for example, ornithine, the sum of citrulline and arginine, GABA, *β*-sitosterol, etc.) was also significant.

##### The Effect of the Accession (Genotype) on the Metabolite Content Variability of Mung Bean Sprouts

A one-way analysis of variance (ANOVA) was carried out to verify the statistical significance of the associations between variability in the content of biochemical compounds in sprouts and accession (genotype) to which the tested sprouts belong. The accession (genotype) significantly influenced the variability in the content of citric acid (the effect size percentage was 21.7%); nicotinic acid (25.7); 4-hydroxycinnamic acid (22.5); ribonic acid (29.2); erythronic acid (15.4); valine (35.0); leucine + isoleucine (33.0); tryptophan (26.1); serine (38.2); tyrosine (25.0); oxoproline (52.7); sorbitol (33.1); methyl-inositol (35.0); ononitol (33.0); chiro-inositol (32.3); rhamnose (31.0); glucose derivatives (56.1); rutinose and derivatives (31.3); campesterol (23.5); stigmasterol (37.5); arachidic acid (29.9); C18:3 (29.2); acylglycerols (27.9); and adenosine(25.3) (Appendix A). Metabolites whose variation depended on the accession were much less numerous than those depending on the germination conditions.

##### The Effect of the Interaction (Environment × Genotype) on the Metabolite Content Variability of Mung Bean Sprouts

The variability of only 12 compounds out of the 116 identified in mung bean sprouts leaves depended both on the conditions of germination and on the genotype (accession): citric acid, 4-hydroxycinamic acid, ribonic acid, erythronic acid, tryptophan, tyrosine, sorbitol, methyl-inositol, ononitol, rutinose and derivatives, stigmasterol, and acylglycerols. The variability of these compounds was analyzed using a Factorial ANOVA (Appendix A). The effect of the accession (genotype), germination conditions, and their interaction (environment × genotype) were analyzed. The variability of many compounds was strongly affected by the environment × genotype interaction: the effect size percentage was 2.8 to 29.2%. This factor most strongly influenced the content of citric acid (29.2%), rutinose, and derivatives of the latter (28.1%). The variability of ribonic acid, erythronic acid, tryptophan, methyl-inositol, and acylglycerols was not significantly associated with the environment × genotype interaction. The effect of the accession (genotype) on these 12 metabolites was in the range of 25.0–37.5%, while the environment factor (germination conditions) varied within 4.5–37.7%. Germination conditions most strongly affected citric acid (37.7%), sorbitol (34.2), tryptophan (27.1), ononitol (25.2), and rutinose and its derivatives (24.7). A relatively high effect of the attribution of sprouts to this or that genotype (accession) on the variability of a number of compounds makes it possible to select genotypes with high or low levels of such compounds for breeding purposes—for example, the content of methyl-inositol (the effect size is 35.0%), ononitol (33.0), sorbitol (33.1), stigmasterol (37.5), etc.

It should be noted that the accumulation of citric acid, 4-hydroxycinamic acid, tryptophan, and sorbitol in the leaves of mung bean sprouts is most significantly influenced by the growth conditions (environment), while that of ribonic acid, erythronic acid, tyrosine, methyl-inositol, ononitol, and rutinose with its derivatives is most significantly influenced by the genotype of the accession (Appendix A). It means that primary metabolism compounds (carbohydrates) are the most variable, although there are signs of the impact on compounds involved in the processes of secondary metabolism (4-hydroxycinnamic acid, and tyrosine).

#### 2.2.2. Factor Analysis Results

A principal factor analysis (PFA) was applied to clarify the variability and structure of relationships among the studied characters.

##### Results of the Factor Analysis of Characters Calculated According to the Content (ppm) of the Identified Metabolites

The analysis outlined three factors for the main part of variances among the characters. The percentage of their variance was 50.4% (Figure 7, Appendix A).

The factor F1 (27.4% of variance) incorporated the interrelations of glyceric acid, total lactones of threonic and erythronic acids, erythronic acid, succinic acid, threonic acid, lactic acid, glycerol, myo-inositol, mannose, glucose, sucrose, *β*-sitosterol, C16:0, C18:0, and C18:2 and the compounds negatively correlated with them: GABA, threonine, phenylalanine, histidine, lysine, glutamic acid, citrulline + arginine, *β*-alanine, ornithine, ornithine lactam, sorbitol, phytol, and urea (Figure 7, Appendix A). The factor F2 correlated (14.0% of variance) C18:3, C24:0, citric acid, malic acid, fumaric acid, benzoic acid, pipecolic acid, phosphate, methyl phosphate, tyrosine, arabinitol, glycerol-3-phosphate, ononitol, and campesterol. The third factor (9.0% of variance) integrated shikimic acid and the compounds negatively correlated with it: erythronic acid, nicotinic acid, 4-hydroxycinnamic acid, oxoproline, MeC18:3, valine, and leucine.

1—lactic acid; 2—pyruvic acid; 3—oxalic acid; 4—succinic acid; 5—malic acid; 6—shikimic acid; 7—citric acid; 8—quinic acid; 9—tartaric acid; 10—fumaric acid; 11—3-hydroxypropionic acid; 12—caffeic acid; 13—2,3-dihydroxybenzoic acid; 14—benzoic acid; 15—nicotinic acid; 16—4-hydroxycinnamic acid; 17—maleic acid; 18—azelaic acid; 19—aconitic acid; 20—dehydroabietic acid; 21—methylmalonic acid; 22—citraconic acid; 23—mesoxalic acid; 24—saccharic acid; 25—glyceric acid; 26—methylglyceric acid; 27—ribonic acid; 28—erythronic acid; 29—threonic acid; 30—total lactones of threonic and erythronic acids; 31—gluconic acid; 32—6-phosphogluconic acid; 33—2-ketogluconic acid; 34—phosphoric acid; 35—phosphate + methyl phosphate; 36—valine; 37—leucine + isoleucine; 38—methionine; 39—threonine; 40—phenylalanine; 41—tryptophan; 42—α-alanine; 43—glycine; 44—serine; 45—tyrosine; 46—histidine + lysine; 47—aspartic acid; 48—asparagine; 49—glutamic acid; 50—glutamine; 51—proline; 52—citrulline + arginine; 53—*β*-alanine; 54—GABA; 55—oxoproline; 56—hydroxyproline; 57—ornithine; 58—pipecolic acid; 59—5-hydroxypipecolic acid; 60—*β*-phenyl-*α*-alanine; 61—ornithine lactam; 62—ethanolamine; 63—glycerol; 64—threitol + erythritol; 65—arabinitol; 66—glycerol-3-phosphate; 67—deoxyglucitol; 68—sorbitol; 69—allo-inositol; 70—methyl-inositol; 71—ononitol; 72—chiro-inositol; 73—myo-inositol; 74—galactinols; 75—phytol; 76—*α*-tocopherol; 77—pentoses; 78—rhamnose; 79—fructose; 80—mannose + galactose; 81—glucose; 82—glucose derivatives; 83—sorbose; 84—sucrose; 85—rutinose and derivatives; 86—raffinose, etc.; 87—campesterol; 88—stigmasterol; 89—*β*-sitosterol; 90—sterol 486; 91—isofucosterol; 92—C9:0; 93—C11:0; 94—C13:0; 95—C16:0; 96—C18:0; 97—C20:0; 98—C22:0; 99—C24:0; 100—C26:0; 101—C18:1; 102—C18:2; 103—C18:3; 104—MeC18:3; 105—OH18:0; 106—OH20:0; 107—OH22:0; 108—OH24:0; 109—OH26:0; 110—OH28:0; 111—acylglycerols; 112—pyrogallol; 113—urea; 114—adenosine; 115—uridine; 116—antirrinoside.

The distribution of the sprout samples in the space of factors F1 and F2 shows that they were expressly divided by the factor F1, forming two groups (Figure 8). The first group included the sprouts exposed to light, and the second was those grown in the dark. Sprouts of the second group were characterized by high levels of glyceric acid, total lactones of threonic and erythronic acids, succinic acid, threonic acid, lactic acid, glycerol, myo-inositol, mannose, glucose, sucrose, *β*-sitosterol, C16:0, C18:0, and C18:2. Sprouts in the first group demonstrated low levels of the abovementioned compounds but, unlike the samples in the first group, they had higher contents of GABA, threonine, phenylalanine, histidine, lysine, glutamic acid, citrulline and arginine, *β*-alanine, ornithine, ornithine lactam, sorbitol, phytol, and urea.

The factor F2 showed specific features of different sprouts in the content of C18:3, C24:0, citric, malic, fumaric, benzoic and pipecolic acids, phosphate, methyl phosphate, tyrosine, arabinitol, glycerol-3-phosphate, ononitol, and campesterol. Sprouts with low levels of these compounds are located in the lower part of the figure, and those with higher levels are in the upper part. It should be pointed out that the sprouts belonging to the same accession are located close to each other. So, the factor F2 made it possible to differentiate the samples and identify accession-specific properties.

The sprouts of k-14407 (No. 6–10) had the lowest content of all the second factor’s compounds under the light; in the dark, the levels of these compounds were medium or high (No. 1–5). Contrariwise, the light-exposed sprouts of k-14408 (No. 21–25) were located in the area characterized by the highest values, while the dark-exposed ones (No. 11–15) were in the area of medium values. Mung bean sprouts of k-14416 manifested medium values in the light (No. 26–30) and in the dark (No. 16–20), i.e., the levels of the second factor’s compounds in sprouts were not affected by the conditions of germination.

In the space of factors F1 and F3, the sprouts were divided only by factor F1, differentiating them into groups associated with the exposure pattern (dark and light). Factor F3 revealed only individual features in the content of shikimic, erythronic, nicotinic and 4-hydroxycoric acids, oxoproline, MeC18:3, valine, and leucine in different sprouts. In Appendix A, sprouts with high levels of shikimic acid and low levels of erythronic, nicotinic and 4-hydroxycoric acids, oxoproline, MeC18:3, valine, and leucine are located in the lower part, while those with low levels of shikimic acid and high levels of other compounds from this factor are in the upper part. There was no clear differentiation by accessions.

##### Results of the Factor Analysis of Characters Calculated According to the Relative Content (%) of Metabolites

The factor analysis of characters calculated as a percentage of individual compounds in the total amount of each of the identified classes of compounds (total organic acids, amino acids, saccharides, alcohols, lipids, and phytosterols) revealed several factors (Figure 9 and Figure 10, Appendix A). They determined 47.2% of the total variability of characters. The factor F1 (28.1% of variance) included myo-inositol, glyceric acid, serine, threonic acid, total lactones of threonic and erythronic acids, leucine, *α*-alanine, proline, aspartic acid, sucrose, and the compounds negatively correlated with them: methyl-inositol, *β*-sitosterol, sorbitol, citric acid, stigmasterol, malic acid, C28:0, phenylalanine, citrulline + arginine, ornithine, deoxyglucitol, ononitol, chiro-inositol, phytol, glucose + derivatives, and sorbose.

The factor F2 (10.5%) harbored saturated fatty acids, lactic acid, ethanolamine, C22:0, and the compounds negatively correlated with them: unsaturated fatty acids, campesterol, tryptophan, and C18:2. The factor F3 (8.7%) combined shikimic acid, glycerol, arabinitol, and the compounds negatively correlated with them: erythronic and 3-hydroxypropionic acids.

The results of the study into the proportions of compounds differed from the abovementioned analysis where the factors were based on the content of all identified compounds. Here, the factor F1 also clearly divided the sprouts according to growth conditions (Figure 10). However, it was myo-inositol, not glyceric acid, that was the leading character determining the coordinated variability of the others. All light-exposed sprouts demonstrated high content of myo-inositol and the compounds positively correlated with it. They are located on the left side of the scatterplot, and dark-exposed ones with low values are on the right. Within the factor F2, where saturated and unsaturated FAs and campesterol were responsible for the variability of characters, the sprouts were distributed in accordance with their individual characteristics. It is interesting to note that the light-exposed plant samples of the same accession were grouped close to each other, while such differentiation was not observed for the dark-exposed ones.

1—lactic acid; 2—pyruvic acid; 3—oxalic acid; 4—succinic acid; 5—malic acid; 6—shikimic acid; 7—citric acid; 8—quinic acid; 9—tartaric acid; 10—fumaric acid; 11—3-hydroxypropionic acid; 12—caffeic acid; 13—2,3-dihydroxybenzoic acid; 14—benzoic acid; 15—nicotinic acid; 16—4-hydroxycinnamic acid; 17—maleic acid; 18—azelaic acid; 19—aconitic acid; 20—dehydroabietic acid; 21—methylmalonic acid; 22—citraconic acid; 23—mesoxalic acid; 24—saccharic acid; 25—glyceric acid; 26—methylglyceric acid; 27—ribonic acid; 28—erythronic acid; 29—threonic acid; 30—total lactones of threonic and erythronic acids; 31—gluconic acid; 32—6-phosphogluconic acid; 33—2-ketogluconic acid; 34—valine; 35—leucine + isoleucine; 36—threonine; 37—methionine; 38—phenylalanine; 39—tryptophan; 40—histidine + lysine; 41—citrulline + arginine; 42—α-alanine; 43—glycine; 44—proline; 45—serine; 46—tyrosine; 47—aspartic acid; 48—asparagine; 49—glutamic acid; 50—glutamine; 51—*β*-alanine; 52—GABA; 53—hydroxyproline; 54—oxoproline; 55—*β*-phenyl-*α*-alanine; 56—pipecolic acid; 57—5-hydroxypipecolic acid; 58—ornithine; 59—ornithine lactam; 60—ethanolamine; 61—glycerol; 62—threitol + erythritol; 63—arabinitol; 64—glycerol-3-phosphate; 65—deoxyglucitol; 66—sorbitol; 67—allo-inositol; 68—methyl-inositol; 69—ononitol; 70—chiro-inositol; 71—myo-inositol; 72—galactinols; 73—phytol; 74—*α*-tocopherol; 75—fatty acids; 76—acylglycerols; 77—saturated fatty acids; 78—unsaturated fatty acids; 79—campesterol; 80—stigmasterol; 81—*β*-sitosterol; 82—sterol 486; 83—isofucosterol; 84—pentoses (arabinose, ribose, and xylose); 85—rhamnose; 86—fructose; 87—mannose + galactose; 88–glucose + derivatives; 89—sorbose; 90—sucrose; 91—rutinose + derivatives; 92—raffinose, etc.; 93—C9:0; 94—C11:0; 95—C13:0; 96—C16:0; 97—C18:0; 98—C20:0; 99—C22:0; 100—C24:0; 101—C26:0; 102—C28:0; 103—C18:1; 104—C18:2; 105—C18:3.

In the space of factors F1 and F3, the sprouts were clearly divided only by factor F1, according to the growth conditions (Appendix A). Factor F3, associated with the content of shikimic, erythronic and 3-hydroxypropionic acids, glycerol, and arabinitol, did not show a clear differentiation of sprouts according to these characteristics. The sprouts are arranged on the graph according to their individual properties: sprouts with a high content of shikimic acid are in the upper part, and those with high levels of erythronic and 3-hydroxypropionic acids, glycerol, and arabinitol are in the lower part.

### 2.3. Nutritional Value of the Analyzed Mung Bean Sprouts

The results of the study showed that the leaves of mung bean sprouts, grown in the dark and in the light, contained a fairly wide range of health-friendly metabolites in the free form, including bioactive ones. The highest amounts were recorded for the classes of metabolites participating in human protein, carbohydrate, and energy metabolisms: amino acids, saccharides, sugar alcohols, and organic acids.

Both groups of sprouts (DS and LS) contained all the essential amino acids, whose shares were 36% (DS) and 28% (LS) (Appendix A) of the total amino acid content. It is known that amino acids, in addition to protein metabolism, are involved in metabolic processes of other compounds. Phenylalanine plays an essential role in the synthesis of phenolic compounds, one of the main classes of antioxidants. Glutamic acid is part of another antioxidant, glutathione tripeptide, and along with *γ*-aminobutyric acid, is a neurotransmitter in the human central nervous system. Histidine is part of hemoglobin and a precursor of histamine, a neurotransmitter and mediator of inflammations. The biosynthesis of catecholamines and adrenaline is associated with tyrosine, and the biosynthesis of nucleotides is associated with glutamine. *β*-alanine is part of pantothenic acid, a component of coenzyme A—one of the key metabolites [58,73].

Saccharides were present in the leaves of the sprouts in an easily digestible form—as di- and monosaccharides. Raffinose, a representative of *α*-oligogalactosides—compounds associated with the phenomenon of flatulence—was identified in singular sprouts. Metabolism of monosaccharides is linked to metabolic processes of aldonic acids and sugar alcohols, whose content in the tested sprouts was quite significant. Myo-inositol, a sugar alcohol, plays an important role in human metabolism. Myo-inositol and its derivatives interact with specific inositol-dependent proteins, participating in intracellular signal transmission. In this way myo-inositol is involved in maintaining normal functions of various human systems: cardiovascular system, immunity, reproductive functions, connective tissue structure, central nervous system, sugar metabolism, kidney and liver functions, etc. Diverse bioactive properties of myo-inositol and its derivatives make it possible to recognize this metabolite as a promising pharmaceutical component [50,74]. Phytic acid, a derivative of myo-inositol, is reported to have a negative effect on the absorption of proteins and minerals, so it is considered an antinutrient. However, as mentioned above, there is a significant decrease in its content during germination of mung bean seeds, and we did not detect it.

The identified organic acids participate in the basic energy processes of metabolism, are involved in the biosynthesis of other groups of metabolites, and normalize the functioning of the digestive system. Succinic acid, a metabolite of the Krebs cycle, can also act as an endocrine stimulus, demonstrates antioxidant activity, and plays a role in the correction of mitochondrial dysfunctions associated with the pathogenesis of a number of diseases, including atherosclerosis, diabetes mellitus, neurodegenerative diseases, and cancer [75]. Lactic acid is important for brain metabolism, which is especially vital in overcoming the consequences of traumatic brain injuries. Lactic acid can influence the reduction of lipolysis in adipose tissues and calcium bioavailability processes, and has anti-inflammatory properties [76,77].

Mung bean leaves contain a significant amount of phosphoric acid; phosphorus is also part of a number of metabolites. This element plays an important role in maintaining human health [78].

The fatty acid composition of the sprouts was well balanced. Phytosterols present in mung bean leaves were previously considered as capable of bringing down cholesterol levels in human blood. Recent studies, however, have expanded the understanding of the possibilities of their bioactivity: diets rich in phytosterols contribute to the prevention of cardiovascular diseases, a number of cancers, liver disorders, and intestinal diseases and have a positive effect on the functioning of the immune system [79,80].

Caffeic acid and its derivatives, representing the class of phenolic compounds, possess a wide range of pharmacological properties, such as antioxidant, anti-inflammatory, antitumor, antiviral, anticarcinogenic, antitoxic, and neuroprotective ones [81].

Most of the metabolites identified in mung bean leaves have a low molecular weight, which, along with their presence in the sprouts in the free state, helps to improve their digestibility and their inclusion into human metabolic processes.

## 3. Materials and Methods

### 3.1. Materials

Seeds of three mung bean accessions from the VIR collection (k-14407, k-14408, and k-14407 from Kenya) were reproduced in 2019 at VIR Astrakhan Experiment Stations (46°0700 N, 41°0100 E). The climate of the Astrakhan region is dry and sharply continental. The soils in the experimental field were alluvial-meadow, heavy loam, and slightly saline (chloride-sulfate type of salinity). The sum of air temperatures above 10 °C during the vegetation period reached a cumulative 3491.3 growing degree days (°C). The annual amount of precipitation ranged by 111.4 mm. The sowing was carried out in a moist, warmed-up soil layer, when the average daily air temperature reached 14–16 °C. The method of sowing is wide-row. Seeds were sown manually, the width between rows was 140 cm, and the distance between seeds in the row was 10 cm. The depth of seeding was 3–5 cm. Accessions were cultivated under irrigated conditions: during the growing season, six irrigations were provided by sprinklers with an average of 250–300 m^3^/ha. Seeds were harvested for testing in the full ripeness phase.

### 3.2. Germination of Mung Bean Seeds

The mung bean seeds were germinated according to the guidelines approved for grain legume crops (GOST 12038-84) with some modifications [82]. The seeds were evenly spread out at a distance of 4–5 cm from each other on a tray between wet layers of filter paper. Then, the paper with the seeds was rolled up and placed vertically in a glass cup with a small amount of water at the bottom so that the seeds did not come into contact with water, but the paper always remained moist. During the germination process, water was added as needed. The seeds were germinated in Fisons Fitotron 600 H growth chamber (Loughborough, UK) at temperature of 20° C for 10 days in the dark and then 1 day in light conditions. The light condition was carried out using L36W/765 Cool Daylight (OSRAM, Smolensk, Russia).

The sprouts of the accessions differed in their morphological features (Table 2). On the 10th day of germination, one leaf from each sprout was taken for further analysis. The sprouts with the remaining second leaflets were transferred to the light. A day later, the second leaf from the same sprout was taken for further analysis. The leaves exposed to different illumination patterns belonged to the same plant, which helped to identify variations in their metabolite compositions. Morphological trait parameters of sprouts did not change in one day. Only the leaf color changed. Yellow leaflets of the dark-exposed sprouts turned green under the light. Each accession was represented by 10 leaves (of which 5 were collected in the dark and 5 in the light conditions). The total set of DS or LS was represented by 15 leaves.

### 3.3. Extraction Procedure and Sample Preparation for GC-MS Analysis

The leaf was separated from the sprout with a scalpel, weighed on analytical scales Sartorius (Germany), and placed in an Eppendorf-type test tube (2 mL) with an 80% methanol solution. Methanol solution was added in such a way that the leaf was completely immersed in the solution. Test tube with the leaf was placed for 24 h in a freezer (Sanyo Ultra Low, Bunkyoku, Tokyo, Japan) at −80 °C to destroy the cell walls of leaf blades. The sample was thawed at room temperature, then sonicated for 30 s on Elmasonic S30 H (Elma, Singen, Germany) and centrifuged at 10,000 rpm for 10 min on Centrifuge 5417R (Eppendorf, Hamburg, Germany) at 4 °C. The total supernatant (1.0–1.5 mL) was transferred to chromatographic vial and dried completely on a CentriVap Concentrator (Labconco, Kansas, MI, USA). Then, 20 µL of tricosane solution in pyridine (concentration: 1 µg/µL) was added to the sample as an internal standard, and the result was silylated with 20 µL of N,O-bis(trimethylsilyl)trifluoroacetamide for 15 min at 100 °C.

### 3.4. GC-MS Analysis

The GC–MS the analysis was carried out according to previously research [83]. The sample (1.2 µL) was separated using an HP-5MS capillary column (5% phenyl 95% methylpolysiloxane, 30.0 m, 250.00 µm, 0.25 µm; Agilent Technologies, Palo Alto, CA, USA) on an Agilent 6850 gas chromatograph with a quadrupole mass selective detector (Agilent 5975B VL MSD, Agilent Technologies). Conditions of the analysis: inert gas flow in the column 1.5 mL/min; temperature program from +70 °C up to +320 °C, with heating rate 4 °C/min; evaporator temperature +300 °C, flow division ratio 1:20. The chromatogram was registered in the full ion flow scan mode at 2.0 scans per second. Ionization by electron impact was performed at 70 eV, with the ion source temperature 230 °C. The recording of [83], 13 of 16 chromatogram started after 4 min required for solvent removal and went on for 62 min. Compounds were identified using AMDIS software (Version 2.69, National Institute of Standards and Technology, Gaithersburg, MD, USA, http://www.amdis.net (accessed on 19 July 2010). Libraries used in the process of analysis: NIST 2010 (National Institute of Standards and Technology, Gaithersburg, MD, USA, http://www.nist.gov (accessed on 19 July 2010), and the collections of standard compound mass spectra maintained by St. Petersburg State University and the Komarov Botanical Institute [84,85]. These last two databases were developed as the result of previous standard-based chemical determination performed at St. Petersburg University and the Botanical Institute of the Russian Academy of Sciences. The retention indices (RI) were estimated by the calibration of saturated hydrocarbons with the number of C atoms ranging from 10 to 40. A semi-quantitative assay of the metabolite profiles was performedvia calculation of the total ion peak areas with the internal standard method using UniChrom software (UniChrom TM 5.0.19.1134, New Analytic Systems LLC, Minsk, Belarus, www.unichrom.com (accessed on 16 June 2021).

### 3.5. Statistical Analysis

Statistical data processing was performed using the Statistica 7 software. The one-way ANOVA was applied to search for associations between growth conditions, accessions (genotype), and the content of biochemical compounds in sprouts. When carrying out the analysis of variance, all the necessary requirements were met: normality of the distribution among the values of the studied trait; equality of variances in the samples being compared; and random and independent sampling. Levene’s test was used to confirm the equality of variance. In addition to estimating the mean, the analysis of variance included the coefficient *η*^2^ (intraclass correlation), indicating what proportion of the total variability was attributed to the factor. The effect size of a factor (*η*^2^, %) was calculated following Fisher’s model and using Formula (1) [86]:(1)$$\eta^{2}=\frac{{SS}_{factor}}{{SS}_{total}}\times 100\%$$
where *η*^2^ is the effect size percentage of a factor, %; *SS_factor_* is the sum of squared deviations for a factor; and *SS_total_* is the total sum of squared deviations.

For metabolites whose variability was affected by germination conditions and attribution to a certain accession (genotype), a complex analysis was performed using Factorial ANOVA [87]. A search was made for statistically significant associations between germination conditions, accession (genotype), their interaction (environment × genotype), and variability of the identified metabolites.

Variability in the structure of interrelations among traits was analyzed using the factor analysis (Principal Factor Analysis, PFA) [87].

## 4. Conclusions

The results of this study showed that the leaves of mung bean sprouts grown in the dark and in the light contain, in the free form, a fairly wide range of metabolites beneficial for human health. In total, more than 100 compounds were identified, including bioactive ones, representing various classes of plant compounds, with alcohols, amino acids, and saccharides dominating in number among them. The analysis of the metabolomic profiles of the leaves collected from the sprouts showed a fairly high intra- and intervarietal variability of the samples in the content of the identified metabolites. The principal factor analysis divided the sprouts grown in the dark and under the light into two separate groups. This is the evidence of specific features in the metabolism of plants grown under different illumination conditions. The plants grown under the light and those grown in darkness differed in the qualitative and quantitative content of the identified compounds. Variability of metabolite content was more pronounced in the dark-exposed sprouts (DS) than in the light-exposed ones (LS). DS exceeded LS in the content of saccharides, fatty acids, acylglycerols, and phenolic compounds, while LS had a higher level of amino acids. Probably, the composition of metabolites in LS was also affected by the stress (abrupt change of germination conditions) experienced by the sprouts during the transition from dark to light. The analysis of variance demonstrated a statistically significant effect of germination conditions, genotype (accession), and their interaction (environment × genotype) on the content metabolite variability in the sprout leaves within a very short interval (1 day). Preliminary results obtained in this study are promising for further in-depth research of the effect of the genotype on the metabolism of mung bean various compounds.

The results of the present study offer additional information about the effect of light on the composition of nutrients in mung bean sprouts, which is important for their use in various diets and in the development of germination technologies to obtain products with a desired biochemical composition.

## Figures and Tables

**Figure 1 plants-12-03772-f001:**
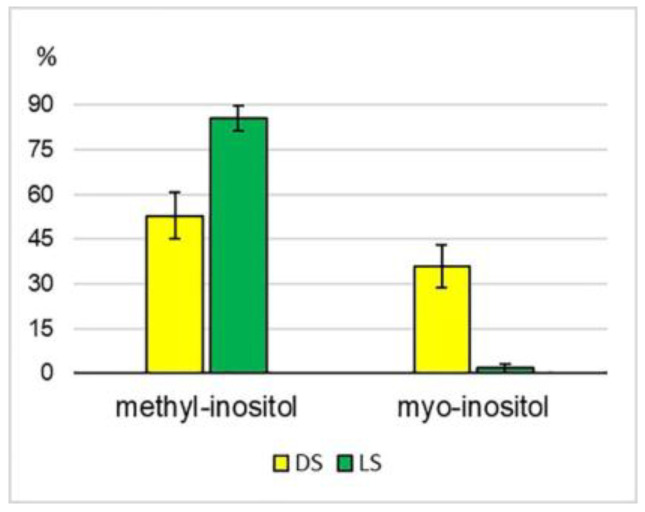
The relative content of main inositol derivatives with reference to the total alcohol content, %.

**Figure 2 plants-12-03772-f002:**
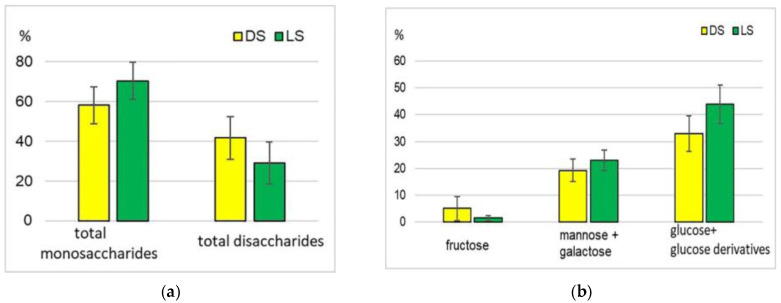
The relative total content of mono- and disaccharides (**a**) and some hexoses (**b**), with reference to the total saccharide content, %.

**Figure 3 plants-12-03772-f003:**
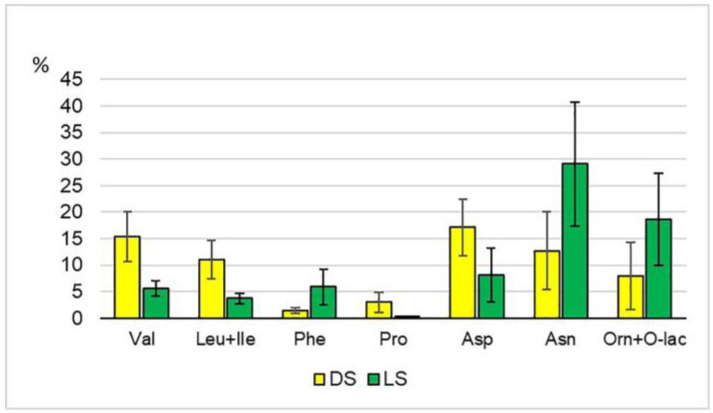
The relative content of some amino acids with reference to their total amount, %. Val—valine; Leu + Ile—leucine + isoleucine; Phe—phenylalanine; Pro—proline; Asp—aspartic acid; Asn—asparagine; Orn + O–lac—ornithine + ornithine lactam.

**Figure 4 plants-12-03772-f004:**
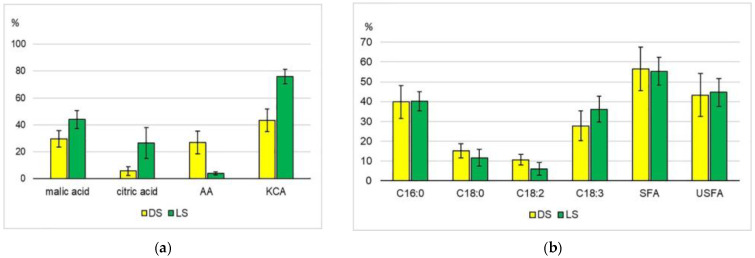
The relative content of some organic acids (**a**) and major free fatty acids (**b**) with reference to their total amount, %. AA—aldonic acids; KCA—Krebs cycle acids; SFA—saturated FA; USFA—unsaturated FA; C16:0—palmitic acid; C18:0—stearic acid; C18:2—linoleic acid; C18:3—linolenic acid.

**Figure 5 plants-12-03772-f005:**
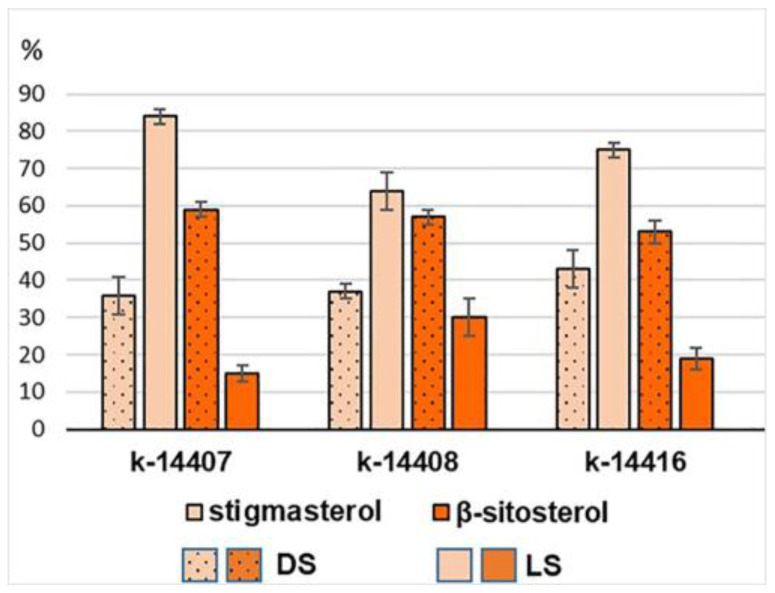
The relative content of major phytosterols with reference to their total amount in the sprout accessions.

**Figure 6 plants-12-03772-f006:**
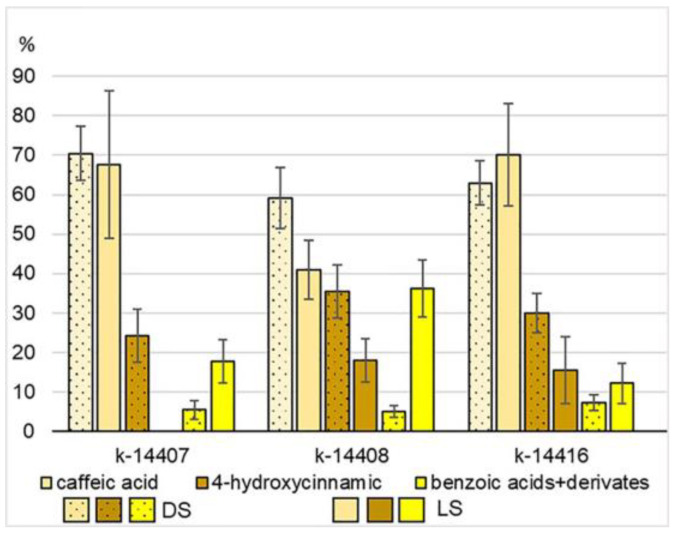
The relative content of major phenolic acids with reference to phenolic compounds total amount in the sprout accessions.

**Figure 7 plants-12-03772-f007:**
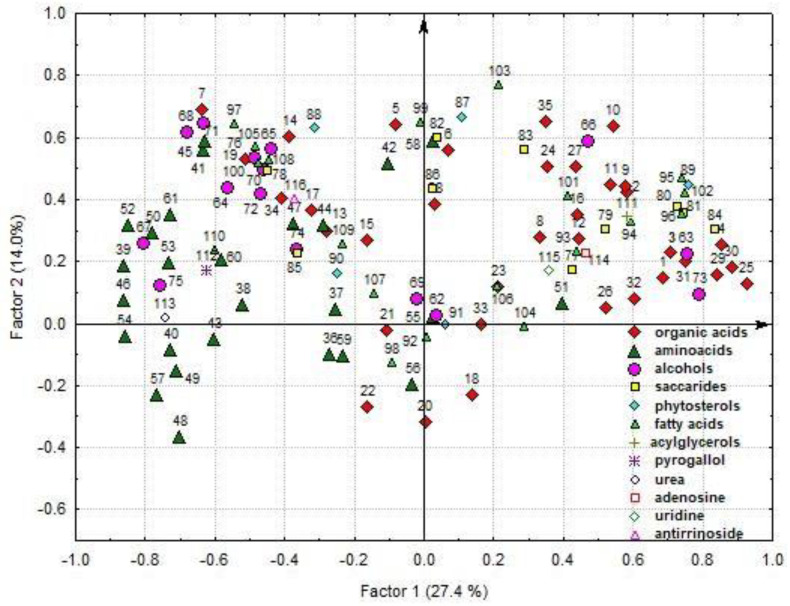
Factor loadings on the characters. A scatterplot of the identified compounds in the space of the first and second factors.

**Figure 8 plants-12-03772-f008:**
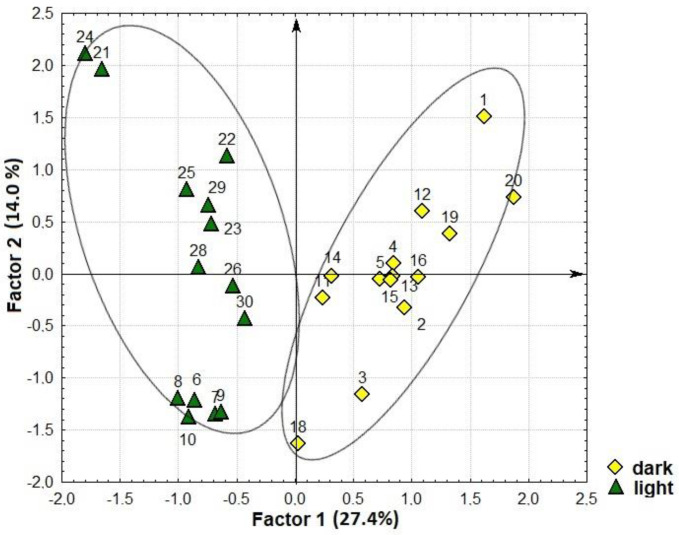
A scatterplot of mung bean sprouts in the space of the first and second factors, calculated according to the content of all identified compounds.

**Figure 9 plants-12-03772-f009:**
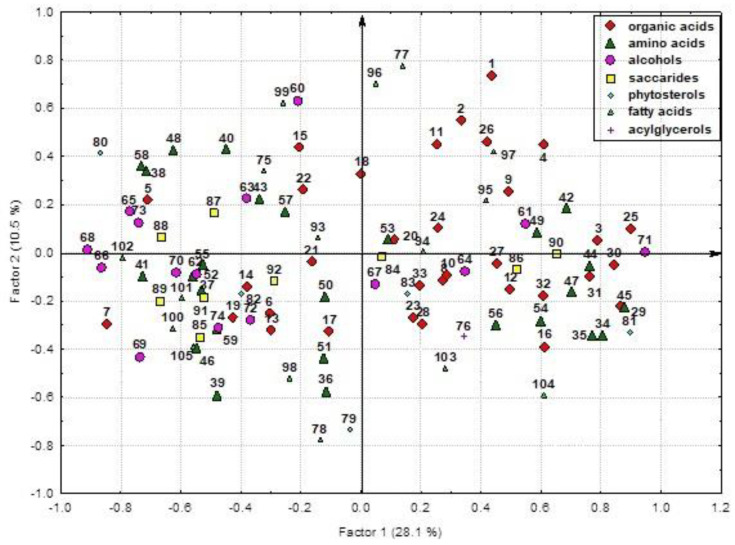
Factor loadings on the characters calculated as percent proportions of individual compounds in the total amount of each identified class of compounds (total organic acids, amino acids, saccharides, alcohols, lipids, and phytosterols).

**Figure 10 plants-12-03772-f010:**
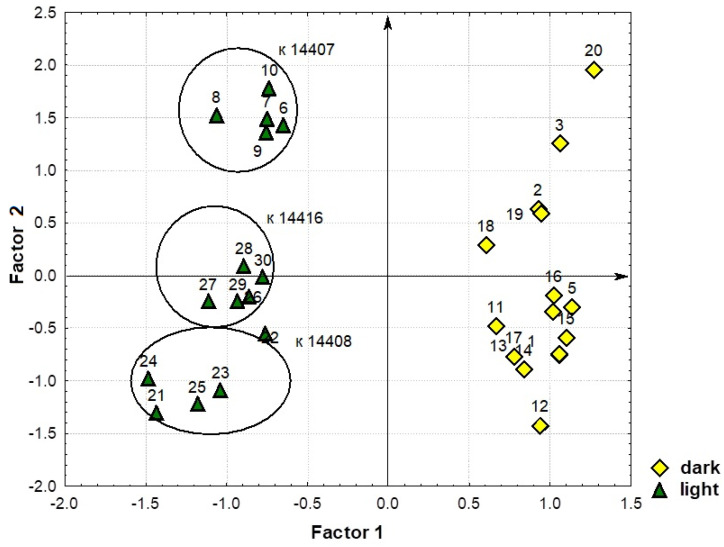
A scatterplot of mung bean sprouts in the space of the first and second factors, calculated according to the proportions of individual substances (%) in the total amount of each identified class of compounds.

**Table 1 plants-12-03772-t001:** The total content of major metabolites in the leaves of mung bean sprouts (ppm).

Accession	k-14407		k-14408		k-14416		Total for the Group	
	Dark	Light	Dark	Light	Dark	Light	DS **	LS ***
Alcohols	4268 ± 2544 *	4178 ± 1112	8615 ± 2330	8311 ± 2464	5788 ± 2887	6658 ± 1719	6255 ± 3052	6362 ± 2501
Total saccharides	6101 ± 4019	1168 ± 441	5895 ± 2031	2522 ± 183	8584 ± 6215	2586 ± 376	6737 ± 4078	2057 ± 759
Amino acids	1906 ± 742	8171 ± 1418	3166 ± 1579	6384 ± 2468	1296 ± 922	5826 ± 2263	2181 ± 1332	6863 ± 2180
Organic acids	1715 ± 969	740 ± 141	1694 ± 478	1991 ± 180	2210 ± 1744	2532 ± 1861	1873 ± 1124	1754 ± 1268
Fatty acids	1199 ± 567	517 ± 106	1059 ± 91	936 ± 174	1299 ± 551	844 ± 186	1178 ± 426	759 ± 241
Acylglycerols	1391 ± 114	15 ± 10	806 ± 437	680 ± 141	2431 ± 2070	1408 ± 431	1479 ± 1228	650 ± 617
Phytosterols	293 ± 111	108 ± 11	312 ± 24	315 ± 35	305 ± 120	300 ± 41	303 ± 86	237 ± 104
Phenolic compounds	91 ± 20	25 ± 5	190 ± 172	120 ± 29	120 ± 53	109 ± 23	135 ± 109	83 ± 49
Phosphoric acids	1660 ± 462	1384 ± 148	1711 ± 219	2393 ± 884	1281 ± 223	1962 ± 841	1570 ± 358	1909 ± 779

* Mean value ± standard deviation; ** DS—dark-exposed sprouts; *** LS—light-exposed sprouts.

**Table 2 plants-12-03772-t002:** Morphological description of the ten-day-old mung bean sprouts.

Accession (VIR Catalogue No.)	Stem Length, cm	Root Length, cm	Stem Diameter, mm	First Leaflet Length, cm
k-14407	10.6–11.3	15.8–17.5	2.0–3.0	1.9–2.6
k-14408	12.6–13.2	15.9–17.1	2.0–3.0	2.4–2.9
k-14416	7.4– 9.1	11.5–12.3	1.5–2.0	1.4–1.8

## Data Availability

The data presented in this study are available in Appendix A.

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
