# Peer review of "The Effect of Illumination Patterns during Mung Bean Seed Germination on the Metabolite Composition of the Sprouts"

_plants, 2023, doi:10.3390/plants12213772_

Round 1
Reviewer 1 Report
Comments and Suggestions for Authors
The manuscript describes a study on the influence of light during germination on the metabolite composition of Mung bean (Vigna radiata (L.) Wilczek) sprouts. The study is focused only on metabolites with low molecular weight due to the limitation of GC-MS analysis.
The "Materials and Methods" section should be given in a more structured way.
The sample preparation and GC-MS methods should be described briefly.
The information on the spectral composition and the luminosity of the light source should be given in this section, as well.
Author Response
Dear Reviewer, thank you very much for a detailed analysis of our article.
All changes in the article are marked in yellow.
These are the responses to your comments.
The "Materials and Methods" section should be given in a more structured way.
The sample preparation and GC-MS methods should be described briefly.
The information on the spectral composition and the luminosity of the light source should be given in this section, as well.
In the corrected version of the article, the section Materials and methods is presented in a more structured form. It includes:
3.1 Materials;
3.2 Germination of mung bean seeds;
3.3 Exstraction procedure and sample preparation for GC-MS analysis;
3.4 GC-MS analysis;
3.5 Statistical analysis.
Reviewer 2 Report
Comments and Suggestions for Authors
Line 125 Include in the figure description the meaning of LS and DS. Do the same for all figures where LS and DS are mentioned.
Line 318. This figure is confusing. Authors say that one bar corresponds to stigmasterol and the other to b-sitosterol, differentiated by colors, and some are for sprouts in dark and light, differentiated by pattern, then what do the last bars represent, the ones in darker red color?
Line 345 Figure 7 is a similar case to Figure 6. These figures need more explanation.
Figure 8, Authors should include the % variation explained by each factor in the graph. According to the text, only the 50.4 % variation is explained by three factors, however, the authors presented a two-dimensional graph. First, the variation explained seems to be low, and why did they dismiss the third factor? Moreover, looking at Figure 8, it seems that it is not useful due to there are no separation or classification among treatments,
Figure 9, similar to the previous figure, needs the percentage of variation in each factor.
Consider if preserving Figure 10 contributes to important information.
I suggest to keep figures 9 and 11.
The references section must be reviewed.
Names of plants or bacteria must be in italics.
Comments on the Quality of English LanguageThe English language is fine, but I am not a native speaker, probably the manuscript needs a complete review by an English native speaker.
Author Response
Dear Reviewer, thank you very much for a detailed analysis of our article.
All changes in the article are marked in yellow.
These are the responses to your comments.
Line 125 Include in the figure description the meaning of LS and DS. Do the same for all figures where LS and DS are mentioned.
Line 318. This figure is confusing. Authors say that one bar corresponds to stigmasterol and the other to b-sitosterol, differentiated by colors, and some are for sprouts in dark and light, differentiated by pattern, then what do the last bars represent, the ones in darker red color?
Line 345 Figure 7 is a similar case to Figure 6. These figures need more explanation.
In all figures, the designations of DS and LS sprouts are included.
Figures 6 and 7 of the first version of the article and the captions to them were slightly changed. In the current version of the article figure1 was deleted, therefore, the numbering of the figures has changed. In the current version figures mentioned above are figures 5 and 6.
In our study, we used principal factors analysis– PFA, not PCA. In most cases, these two methods usually yield very similar results. However, principal components analysis is often preferred as a method for data reduction, while principal factors analysis is often preferred when the goal of the analysis is to detect structure. Factor analysis is a statistical procedure to identify interrelationships that exist among a large number of variables, i.e., to identify how suites of variables are related. Therefore, in our study, we used factor analysis.
Figure 8, Authors should include the % variation explained by each factor in the graph.
We have added the % variation (FD, %) explained by each factor in the graph.
According to the text, only the 50.4 % variation is explained by three factors, however, the authors presented a two-dimensional graph. First, the variation explained seems to be low, and why did they dismiss the third factor?
We did not consider it expedient to build a three-dimensional graph. We also plotted the arrangement of the samples in the space of the first and third factors, placing the graphs in the Supplement. Figure S1. A scatterplot of mung bean sprouts in the space of the first and third factors, calculated according to the content of all identified compounds. Figure S2. A scatterplot of mung bean sprouts in the space of the first and third factors, calculated according to the proportions of individual compounds (%) in the total amount of each identified class of compounds). A description of these graphs was added to the article.
To provide better understanding of all the graphs (Figure 8, Figure 10, Figure S1, and Figure S2), we included Table S8 (Factor loadings on the characters (identified compounds) of mung bean sprouts) and Table S9 [Factor loadings on the characters (calculated as percentages of individual compounds in the total amount of each identified class of compounds (total amount of organic acids, amino acids, saccharides, alcohols, lipids, and phytosterols)] in the Supplement.
Moreover, looking at Figure 8, it seems that it is not useful due to there are no separation or classification among treatments,
Figure 9, similar to the previous figure, needs the percentage of variation in each factor.
Consider if preserving Figure 10 contributes to important information.
I suggest to keep figures 9 and 11.
Figure 8 (in the current version 7) and Figure 10 (9) are placed within the text of the article, because they are much smaller than the tables with factor loadings on the characters and help to interpret Figures 9 (8) and 11 (10) . Analyzing these figures concurrently with the factor loadings on the characters and the arrangement of samples in the factor space, we are able to characterize any sample and find out in which area it is localized – with high or low content of a particular compound. Therefore, when PFA is used, either a graph or a table with factor loadings on the characters is given. With them, it is much easier to interpret graphs with the arrangement of samples in the factor space.
The references section must be reviewed.
Names of plants or bacteria must be in italics.
The references section was corrected.
Comments on the Quality of English Language
The English language is fine, but I am not a native speaker, probably the manuscript needs a complete review by an English native speaker.
The text of the article was translated by a professional translator of FAO.
Reviewer 3 Report
Comments and Suggestions for Authors
This study showed the detailed chemical composition of the sprouts of three mung bean accessions.
The results are of great importance for mung bean breeders and consumers. This MS was written in a very good way, and it was well-designed.
However, there are some points that need to be revised, as follows:
Please find the attached comments.

Author Response
Dear Reviewer, thank you very much for such a detailed analysis of our article.
These are the responses to your comments.
All changes in the article are marked in yellow.
Major remark.
- I have an issue with the original idea of the manuscript. As well known that dark or
light has an important impact on seed germination and the produced seedlings. The
authors did not explain clearly why the seeds were germinated in the dark for 10 days,
then they were moved to the light for one day. This needs an explanation. It is not clear
why the authors consider light as abiotic stress against the sprouts.
- The objective of the study is not clear, is it from a nutritional or a botanical view.
The authors should stress why they carried out this study.
1,3. In our study, mung bean sprouts are recognized as a product that can be used in various types of human diets (functional, dietetic, therapeutic, etc.) due to their biochemical composition. Besides, those who practice vegetarianism or veganism are active consumers of sprouts. Most of the studies examining the metabolite composition of mung bean sprouts were performed on whole sprouts. However, people also consume certain parts of them – for example, leaves. Studies of the leaf metabolites composition and the effect of various germination conditions on it are extremely small.
Mung bean seeds are germinated under various illumination patterns. In our case we chose germination in the dark The sprouts were grown for 10 days according to germinated according to the guidelines approved for grain legume crops (GOST 12038-84). A full-fledged sprout is formed by the end of this interval, with a well-developed radicle, hypocotyl, epicotyl, a normal apical bud, and two leaflets. At the same time, however, the plant has not yet passed into the next stage of development.
The plant exists as a sprout for a short period of time, and metabolic processes are very intense during this interval. Leaf metabolism is closely associated with photosynthesis, so it seems interesting how quickly the metabolite composition in sprout leaves can change when a sprout is removed from the conditions where no photosynthesis is induced (dark) and placed under the conditions where photosynthesis is active (light).
In our study, we limited ourselves to one day The leaves exposed in different illumination patterns belonged to the same plant, which contributed to the identification of differences in their metabolite compositions.
We do not recognize light as an abiotic stress. It is an abrupt change in of germination conditions (the illumination pattern) i.e., the transition of sprouts transfer from dark to light, that is considered a possible stress factor.
The metabolite compositions of plants grown under various environmental conditions may have their own specific features. At the same time, the metabolite composition in LS can also be affected by the response of a plant (genotype) to its transition to different germination conditions. In view of this, the article discusses the changes in the content of metabolites that are involved, as shown by various researchers, in the response of plants to various stressors. Actually, this discussion is formatted as a hypothesis.
The purpose and objectives of the study were corrected.
- The authors did not show the experimental design they used in their study, as there
are two factors (the accessions, and the growth conditions).
The scheme of the research was described in more detail in the section Materials and methods.
We carried out additional studies using a one-way ANOVA method to determine the reliability of the effect of the accession (genotype) on the variability of the composition of metabolites. We also used the Factorial ANOVA method to identify the effect of the accession *growth conditions interaction. Additional information was added in the appropriate parts of article.
Minor remarks.
- 1. In Table S1: Please check the mean values of the detected compounds. Some mean
values do not make sense such as Methyl-glyceric acid, Rutinose + rutinose derivatives,
Raffinose, Ethanolamine, etc. There is a conflict between the minimum, the maximum,
and the mean values.
Table S1. The tested sprouts differed in the content of compounds, some of which demonstrated a fairy wide range of variability. Besides, some compounds were detected only in singular plants. Such metabolites as methylglyceric acid, raffinose, ethanolamine, etc., were not identified in all of the sprouts. However, when the mean value was calculated, the total (combined) set of DS or LS (15 samples) was taken into account.
- Figure 1 contains the same data in Table 1, I suggest deletion of Figure 1, and deleting
from line 119.
Figure1 was deleted.
- L576-577: this sentence and Table 2 should be moved to the results part.
- In Figure 2: Is this (Morphological description of the ten-day-old mung bean sprouts),
including one-day in the light or only for the dark.
We refer mung bean sprouts to materials because they are the source of the objects of our research - leaves. The table is presented in order to characterize the studied material in more detail.
The morphological features that were measured for 10-day-old sprouts are presented in Table 2. But morphological characters of the sprouts did not change in one day, so the data presented in the table describe both 10-day and 11-day sprouts. Only the leaf color did change. Yellow leaflets of the dark-exposed sprouts turned green under the light. The leaves exposed to different illumination patterns belonged to the same plant, which helped to identify variations in their metabolite compositions
- Please check the legends of Figures 6 and 7.
In the current version of the article figure1 was deleted, therefore, the numbering of the figures has changed. Figures 6 (5), 7(6) and the figure captions have been slightly changed.
- L335-336: 4-Hydroxycinnamic acid
L335-336 were corrected:
In all DS and the LS of k-14408 and k-14416 4-hydroxycinnamic acid content varied within 13–148 ppm and 5–40 ppm, respectively. This metabolite was not identified in the LS of k-14407.
- L359-363 “2.2. Statistical analysis” and “2.2.1. One-way ANOVA results” A oneway analysis …….. in them. This should be moved to the materials and methods part.
- L363-383: Starting from “The illumination pattern …. This part belongs to the
results, and I suggest using this subtitle “Effect of growth conditions or effect of light
conditions ….
- L384-386: This should be moved to the materials and methods part.
Section 2.2. Statistical analysis has been slightly changed.
Materials and methods:
- How did you get these accessions, add the time of collection, and a specific location.
- Please give detailed information on how you carry out the germination experiment
including the growth conditions in both dark and light.
- How about the color of the sprouts after dark and light conditions.
- Please explain the extraction method in detail
Section Materials and methods is presented in a more structured form. It includes:
3.1 Materials;
3.2 Germination of mung bean seeds;
3.3 Exstraction procedure and sample preparation for GC-MS analysis;
3.4 GC-MS analysis;
3.5 Statistical analysis.

Round 2
Reviewer 3 Report
Comments and Suggestions for Authors
All comments were addressed by the authors, and the MS was markedly improved.